

# Dark matter, fine-tuning and $(g-2)_\mu$ in the pMSSM

**Melissa van Beekveld[1]\*, Wim Beenakker[2,3], Marrit Schutten[2,4] and Jeremy de Wit[2]**

**1** Rudolf Peierls Centre for Theoretical Physics, 20 Parks Road,
Oxford OX1 3PU, United Kingdom
**2** THEP, Radboud University, Heyendaalseweg 135,
6525 AJ Nijmegen, the Netherlands
**3** Institute of Physics, University of Amsterdam, Science Park 904,
1018 XE Amsterdam, the Netherlands
**4** Van Swinderen Institute for Particle Physics and Gravity, University of Groningen,
9747 AG Groningen, The Netherlands

\* melissa.vanbeekveld@physics.ox.ac.uk

## Abstract

In this paper we perform for the first time an in-depth analysis of the spectra in the phenomenological supersymmetric Standard Model that simultaneously offer an explanation for the $(g-2)_\mu$ discrepancy $\Delta a_\mu$, result in the right dark-matter relic density $\Omega_{DM}h^2$ and are minimally fine-tuned. The resulting spectra may be obtained from [1]. To discuss the experimental exclusion potential for our models, we analyse the resulting LHC phenomenology as well as the sensitivity of dark-matter direct detection experiments to these spectra. We find that the latter type of experiments with sensitivity to the spin-dependent dark-matter–nucleon scattering cross section $\sigma_{SD,p}$ will probe all of our found solutions.

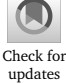

# 1  Introduction

The Large Hadron Collider (LHC) has been searching for over a decade for signs of physics that originate from beyond-the-Standard-Model (BSM) scenarios, including searches for signals that originate from supersymmetric (SUSY) particle production. These high-energy searches are complemented by low-energy experiments such as dark-matter (DM) experiments, or experiments that search for small deviations in known Standard-Model (SM) processes from their SM prediction. In the former category, the XENON1T [2,3], PandaX-II [4,5] and PICO [6–8] experiments provide limits on the DM-nucleus scattering cross section, whereas the Planck collaboration provides a precise measurement of the DM relic abundance [9]. In the latter category, the anomalous magnetic moment of the muon $(g-2)_\mu$ plays an important role. There is a long-standing discrepancy between the experimental result [10–12] and the SM prediction for the muon anomalous magnetic moment. The latter is composed of quantum-electrodynamic, weak, hadronic vacuum-polarization, and hadronic light-by-light contributions, and reads [13–34]

$$a_\mu^{\mathrm{SM}} = \frac{(g-2)_\mu}{2} = 116\,591\,810(43) \times 10^{-11}\,, \tag{1}$$

where the value between parentheses represents the theoretical uncertainty. The improved experimental results obtained at Fermilab [35–38], combined with the Brookhaven result [10–12] read

$$a_\mu^{\mathrm{exp}} = 116\,592\,061(41) \times 10^{-11}\,, \tag{2}$$

showing that the deviation is now

$$\Delta a_\mu = a_\mu^{\mathrm{exp}} - a_\mu^{\mathrm{SM}} = 251(59) \times 10^{-11}\,. \tag{3}$$

An independent experiment with different techniques than those employed by the Fermilab experiment is being constructed at J-PARC [39,40].

The Minimal Supersymmetric Standard Model (MSSM) with $R$-parity conservation predicts a DM candidate and can simultaneously provide an explanation for the $(g-2)_\mu$ discrepancy [1]. Furthermore, the MSSM provides a solution to the fine-tuning (FT) problem in the Higgs sector that any BSM model introduces, even after taking into account the constraints on colored sparticles originating from the LHC. It is clear that for a rich model such as the MSSM, the interplay between the various experimental results is of crucial importance. In this context, several studies have been performed to study a subset of these constraints. For instance, the interplay between the LHC limits and the $(g-2)_\mu$ discrepancy has been studied in e.g. Ref. [42–49]. DM direct detection (DMDD) searches are complementary in regions of the MSSM parameter space where the LHC has little sensitivity, for example in compressed regions. Papers that explore the DM implications of spectra that explain the $(g-2)_\mu$ discrepancy include Refs. [48–53],

---

[1]A simultaneous explanation of the muon and electron anomalous magnetic moments in the MSSM context is provided in Ref. [41].

where the relic density requirement is not always taken into account. Likelihood analyses or global fits, where all experimental data that constrain the MSSM parameter space are taken into account, have been performed in e.g. Ref. [53–59]. The degree of FT in constrained models that explain the $(g-2)_\mu$ discrepancy is studied in [60,61], whereas the role of FT in spectra with the right DM properties is studied in Ref. [62–66].

In this work we perform for the first time a study of the phenomenology of the MSSM that simultaneously accounts for the DM relic abundance and the observed discrepancy of $(g-2)_\mu$, that includes all DMDD and LHC limits, and that constrains the model-parameter space to models that are minimally fine-tuned. The resulting spectra may be obtained from [1]. The paper is structured as follows. In Section 2 we introduce our notation, the muon anomalous magnetic moment and the electroweak fine-tuning measure. In Section 3 we explain the set-up of our analysis. In Section 4 we explore the phenomenology of the viable spectra, and in Section 5 we present our conclusions.

## 2 The muon anomalous magnetic moment and fine-tuning in the pMSSM

Instead of exploring the full MSSM with 105 free parameters, we focus on the phenomenological MSSM (pMSSM) [67], which has 19 free parameters whose boundary conditions are given at the SUSY scale of $\mathcal{O}(1\text{ TeV})$. In this phenomenologically motivated pMSSM one requires that the first and second generation squark and slepton masses are degenerate, that the trilinear couplings of the first and second generation sfermions are set to zero (leaving only those of the third generation, $A_t$, $A_b$ and $A_\tau$), and that no new sources of CP violation are introduced. In addition one assumes that all sfermion mass matrices are diagonal. The sfermion soft-masses are then described by the first and second generation squark masses $m_{\widetilde{Q}_1}$, $m_{\widetilde{u}_R}$ and $m_{\widetilde{d}_R}$, the third generation squark masses $m_{\widetilde{Q}_3}$, $m_{\widetilde{t}_R}$ and $m_{\widetilde{b}_R}$, the first and second generation of slepton masses $m_{\widetilde{L}_1}$ and $m_{\widetilde{e}_R}$, and the third generation of slepton masses $m_{\widetilde{L}_3}$ and $m_{\widetilde{\tau}_R}$. The Higgs sector is described by the ratio of the Higgs vacuum expectation values $\tan\beta$ and the soft Higgs masses $m_{H_u}$ and $m_{H_d}$. Instead of these parameters, it is customary to use the higgsino mass parameter $\mu$ and the mass $m_A$ of the pseudoscalar Higgs boson as free parameters. The gaugino sector consists of the bino ($\widetilde{B}$), wino ($\widetilde{W}$) and gluino with their mass parameters $M_1(=|M_1|)$, $M_2(=|M_2|)$ and $M_3(=|M_3|)$.

As a result of electroweak symmetry breaking (EWSB), the gaugino and the higgsino interaction eigenstates mix into mass eigenstates, called neutralinos and charginos. The neutralinos, denoted by $\widetilde{\chi}_i^0$ with $i=1,\ldots,4$, are the neutral mass eigenstates of the bino, wino and higgsino interaction eigenstates. The neutralinos are ordered by increasing mass, with $\widetilde{\chi}_1^0$ the lightest neutralino. Given the constraints from DMDD experiments on sneutrino DM, we take the lightest neutralino as lightest-supersymmetric particle (LSP), which makes it our DM candidate. Depending on the exact values of $M_1$, $M_2$ and $|\mu|$, this lightest mass eigenstate can be mostly bino-like (if $M_1$ is smallest), wino-like (if $M_2$ is smallest) or higgsino-like (if $|\mu|$ is smallest). The amount of bino, wino and higgsino mixing of the lightest neutralino is given by $N_{11}$, $N_{12}$ and $\sqrt{N_{13}^2 + N_{14}^2}$, where $N_{ij}$ are the entries of the matrix that diagonalizes the neutralino mass matrix. In the basis of $(\widetilde{B}, \widetilde{W}^0, \widetilde{H}_d^0, \widetilde{H}_u^0)$, this mass matrix is given by

$$M_{\widetilde{\chi}^0} = \begin{pmatrix} M_1 & 0 & -c_\beta s_{\theta_W} M_Z & s_\beta s_{\theta_W} M_Z \\ 0 & M_2 & c_\beta c_{\theta_W} M_Z & -s_\beta c_{\theta_W} M_Z \\ -c_\beta s_{\theta_W} M_Z & c_\beta c_{\theta_W} M_Z & 0 & -\mu \\ s_\beta s_{\theta_W} M_Z & -s_\beta c_{\theta_W} M_Z & -\mu & 0 \end{pmatrix}, \tag{4}$$

with $s_x \equiv \sin x$, $c_x \equiv \cos x$, and the ratio of the SM $W$- and $Z$-boson masses being denoted by $\cos\theta_W = M_W/M_Z$.

The charginos, denoted by $\widetilde{\chi}_i^{\pm}$ with $i = 1, 2$, are the charged mass eigenstates of the wino and higgsino interaction eigenstates, with $\widetilde{\chi}_1^{\pm}$ the lightest chargino. In the basis of $(\widetilde{W}^{\pm}, \widetilde{H}_{u/d}^{\pm})$, their mass matrix at tree level reads

$$M_{\widetilde{\chi}^{\pm}} = \begin{pmatrix} M_2 & \sqrt{2}c_{\beta}c_{\theta_W}M_Z \\ \sqrt{2}s_{\beta}c_{\theta_W}M_Z & \mu \end{pmatrix}. \tag{5}$$

The composition of the lightest chargino is predominantly higgsino when $|\mu| < M_2$, predominantly wino when $M_2 < |\mu|$, or a mixture when the two gaugino parameters are close in value.

## 2.1 Electroweak fine-tuning in the pMSSM

The EWSB conditions link $M_Z$ to the input parameters via the minimization of the scalar potential of the Higgs fields. The resulting equation at one loop is [68,69]

$$\frac{M_Z^2}{2} = \frac{m_{H_d}^2 + \Sigma_d^d - (m_{H_u}^2 + \Sigma_u^u)\tan^2\beta}{\tan^2\beta - 1} - \mu^2 , \tag{6}$$

where the two effective potential terms $\Sigma_u^u$ and $\Sigma_d^d$ denote the one-loop corrections to the soft SUSY breaking Higgs masses (explicit expressions are shown in the appendix of Ref. [69]). In order to obtain the observed value of $M_Z = 91.2$ GeV, one needs some degree of cancellation between the SUSY parameters appearing in Eq. (6). If small relative changes in the SUSY parameters will result in a distinctly different value of $M_Z$, the considered spectrum is said to be fine-tuned, as then a large degree of cancellation is needed to obtain the right value of $M_Z$. FT measures aim to quantify this sensitivity of $M_Z$ to the SUSY input parameters.

The electroweak (EW) FT measure [70, 71] is an agnostic approach to the computation of fine-tuning. We take this approach because a generic broken minimal SUSY theory has two relevant energy scales: a high-scale one at which SUSY breaking takes place, and a low-scale one ($M_{\text{SUSY}}$) where the resulting SUSY particle spectrum is situated and the EWSB conditions must be satisfied. We do not know which and how many fundamental parameters exist for a possible high-scale theory. The EW FT measure does not take such underlying high-scale model assumptions into account for its computation. The EW FT measure ($\Delta_{\text{EW}}$) parameterizes how sensitive $M_Z$ is to variations in each of the coefficients $C_i$, which are evaluated at $M_Z$. It is defined as

$$\Delta_{\text{EW}} \equiv \max_i \left| \frac{C_i}{M_Z^2/2} \right|, \tag{7}$$

where the $C_i$ are

$$C_{m_{H_d}} = \frac{m_{H_d}^2}{\tan^2\beta - 1}, \quad C_{m_{H_u}} = \frac{-m_{H_u}^2\tan^2\beta}{\tan^2\beta - 1}, \quad C_{\mu} = -\mu^2,$$

$$C_{\Sigma_d^d} = \frac{\max(\Sigma_d^d)}{\tan^2\beta - 1}, \quad C_{\Sigma_u^u} = \frac{-\max(\Sigma_u^u)\tan^2\beta}{\tan^2\beta - 1}.$$

The tadpole contributions $\Sigma_u^u$ and $\Sigma_d^d$ contain a sum of different contributions. These contributions are computed individually and the maximum contribution is used to compute the $C_{\Sigma_u^u}$ and $C_{\Sigma_d^d}$ coefficients. We will use an upper bound of $\Delta_{\text{EW}} < 100$ (implying no worse than $\mathcal{O}(1\%)$ fine-tuning on the mass of the $Z$-boson) to determine whether a given set of MSSM parameters is fine-tuned, and use the code from Ref. [64] to compute the measure.

Using this measure, one generically finds that minimally fine-tuned scenarios have low values for $|\mu|$, where $\Delta_{EW} = 100$ is reached at $|\mu| \simeq 800$ GeV [64, 66, 70, 72–76]. The masses of the gluino, sbottom, stop and squarks are allowed to get large for models with low $\Delta_{EW}$ [65, 77, 78]. Therefore, we assume that the masses of these sparticles are above 2.5 TeV (for the gluino), above 1.2 TeV (for the stops and bottoms) and above 2 TeV (for the squarks), such that they evade the ATLAS and CMS limits [2].

## 2.2 The muon anomalous magnetic moment

In the pMSSM, one-loop contributions to $a_\mu$ arise from diagrams with a chargino-sneutrino or neutralino-smuon loop [79]. The expressions for these one-loop corrections read [80]

$$\delta a_\mu^{\widetilde{\chi}^0} = \frac{m_\mu}{16\pi^2} \sum_{i=1}^{4} \sum_{m=1}^{2} \left[ -\frac{m_\mu}{12 m_{\widetilde{\mu}_m}^2} \left( |n_{im}^L|^2 + |n_{im}^R|^2 \right) F_1^N \left( \frac{m_{\widetilde{\chi}_i^0}^2}{m_{\widetilde{\mu}_m}^2} \right) \right.$$
$$\left. + \frac{m_{\widetilde{\chi}_i^0}}{3 m_{\widetilde{\mu}_m}^2} \mathrm{Re}\left[ n_{im}^L n_{im}^R \right] F_2^N \left( \frac{m_{\widetilde{\chi}_i^0}^2}{m_{\widetilde{\mu}_m}^2} \right) \right] \tag{8}$$

$$\delta a_\mu^{\widetilde{\chi}^\pm} = \frac{m_\mu}{16\pi^2} \sum_{k=1}^{2} \left[ \frac{m_\mu}{12 m_{\widetilde{\nu}_\mu}^2} \left( |c_k^L|^2 + |c_k^R|^2 \right) F_1^C \left( \frac{m_{\widetilde{\chi}_k^\pm}^2}{m_{\widetilde{\nu}_\mu}^2} \right) + \frac{2 m_{\widetilde{\chi}_k^\pm}}{3 m_{\widetilde{\nu}_\mu}^2} \mathrm{Re}\left[ c_k^L c_k^R \right] F_2^C \left( \frac{m_{\widetilde{\chi}_k^\pm}^2}{m_{\widetilde{\nu}_\mu}^2} \right) \right], \tag{9}$$

with $m_\mu$ the muon mass, $m_{\widetilde{\mu}_m}$ the first or second smuon mass, $m_{\widetilde{\nu}_\mu}$ the muon sneutrino mass, $i$, $m$ and $k$ the indices for the neutralinos, smuons and charginos and the couplings

$$n_{im}^R = \sqrt{2} g_1 N_{i1} X_{m2} + y_\mu N_{i3} X_{m1}, \qquad n_{im}^L = \frac{1}{\sqrt{2}} (g_2 N_{i2} + g_1 N_{i1}) X_{m1}^* - y_\mu N_{i3} X_{m2}^*, \tag{10}$$
$$c_k^R = y_\mu U_{k2}, \qquad c_k^L = -g_2 V_{k1}. \tag{11}$$

The down-type muon Yukawa coupling is denoted by $y_\mu = g_2 m_\mu/(\sqrt{2} M_W \cos\beta)$, and the SU(2) and U(1) gauge couplings are $g_2$ and $g_1$. The matrices $N$ and $U$, $V$ diagonalize the neutralino and chargino mass matrices (Eq. (4), (5)), while the unitary matrix $X$ diagonalizes the smuon mass matrix $M_{\widetilde{\mu}}^2$, which reads for the pMSSM in the $(\widetilde{\mu}_L, \widetilde{\mu}_R)$ basis

$$M_{\widetilde{\mu}}^2 = \begin{pmatrix} m_{\widetilde{L}_1}^2 + \left( s_{\theta_W}^2 - \frac{1}{2} \right) M_Z^2 \cos(2\beta) & -m_\mu \mu \tan\beta \\ -m_\mu \mu \tan\beta & m_{\widetilde{e}_R}^2 - s_{\theta_W}^2 M_Z^2 \cos(2\beta) \end{pmatrix}. \tag{12}$$

The loop functions $F_{1,2}^N$ and $F_{1,2}^C$ can be found in Ref. [80]. They are normalized such that $F_{1,2}^{N,C}(x=1) = 1$, and go to zero for $x \to \infty$.

At two-loop, the numerical values of the various contributions differ considerably. The photonic Barr-Zee diagrams are the source of the largest possible two-loop contribution. Here a Higgs boson and a photon connect to either a chargino or sfermion loop [81] [3].

As one can see in the expressions above, the chargino-sneutrino and neutralino-smuon contributions are controlled by $M_1$, $M_2$, $\tan\beta$ and $\mu$ (through $m_{\widetilde{\chi}_i^0}$ and $m_{\widetilde{\chi}_k^\pm}$), as well as $m_{\widetilde{L}_1}$ and

---

[2] Note that those limits are shown to be significantly less stringent for MSSM spectra with rich sparticle decays, see e.g. Ref. [59].

[3] Two-loop corrections from sfermion loops contribute with a few percent here as well, since we assume heavy squark masses [82, 83].

$m_{\widetilde{e}_R}$ (through $m_{\widetilde{\mu}_m}$ and $m_{\widetilde{\nu}_\mu}$). They are enhanced when $\tan\beta$ grows large and when simultaneously light ($\mathcal{O}(100)$ GeV) neutralinos/charginos and smuons/sneutrinos exist in the sparticle spectrum. The Barr-Zee diagrams are enhanced by large values of $\tan\beta$, small values of $m_A$ and large Higgs-sfermion couplings. In general, the one-loop chargino-sneutrino contribution dominates over the neutralino-slepton contribution [80], unless there is a large smuon left-right mixing induced by a sizable value for $|\mu|$ [84]. These latter spectra will however result in slightly higher FT values, which is a direct consequence of a higher value of $|\mu|$.

## 3 Analysis setup

To create the SUSY spectra we use SOFTSUSY 4.0 [85], the Higgs mass is calculated using FeynHiggs 2.14.2 [86–90], and SUSYHIT [91] is used to calculate the decay of the SUSY and Higgs particles. Vevacious [92–94] is used to check that the models have at least a meta-stable minimum state that has a lifetime that exceeds that of our universe and that this state is not color/charge breaking [4]. We use SUSY-AI [95] and SMODELS [96–100] to determine the LHC exclusion of a model point. LHC cross sections for sparticle production at NLO accuracy are calculated using Prospino [101]. HIGGSBOUNDS 5.1.1 is used to determine whether the SUSY models satisfy the LEP, Tevatron and LHC Higgs constraints [102–109]. MICROMEGAS 5.2.1 [110–115] is used to compute the DM relic density ($\Omega_{\mathrm{DM}}h^2$), the present-day velocity-weighted annihilation cross section ($\langle\sigma v\rangle$) and the spin-dependent and spin-independent dark-matter–nucleon scattering cross sections ($\sigma_{\mathrm{SD,p}}$ and $\sigma_{\mathrm{SI,p}}$). For DM indirect detection we only consider the limit on $\langle\sigma v\rangle$ stemming from the observation of gamma rays originating from dwarf galaxies, which we implement as a hard cut on each of the channels reported on the last page of Ref. [116]. The current constraints on the dark-matter–nucleon scattering cross sections originating from various dark matter direct detection (DMDD) experiments are determined via MICROMEGAS, while future projections of constraints are determined via DDCALC 2.0.0 [117]. Flavor observables are computed with SuperIso 4.1 [118, 119]. The muon anomalous magnetic moment and its theoretical uncertainty are determined including two-loop corrections and $\tan\beta$ resummation with GM2Calc [82, 120–122].

We use the Gaussian particle filter [123] to search the pMSSM parameter space for interesting areas. The lightest SM-like Higgs boson is required to be in the mass range of 122 GeV $\leq m_h \leq$ 128 GeV. Spectra that do not satisfy the LHC bounds on sparticle masses, branching fractions of $B/D$-meson decays, the DMDD, or DM indirect detection bounds are removed. Our spectra are furthermore required to satisfy the LEP limits on the masses of the charginos, light sleptons and staus ($m_{\widetilde{\chi}_1^\pm} > 103.5$ GeV, $m_{\widetilde{l}^\pm} > 90$ GeV and $m_{\widetilde{\tau}^\pm} > 85$ GeV) [124, 125], and the constraints on the invisible and total width of the $Z$-boson ($\Gamma_{Z,\mathrm{inv}} = 499.0 \pm 1.5$ MeV and $\Gamma_Z = 2.4952 \pm 0.0023$ GeV) [126]. The spectra surviving all constraints are available via [1] [5].

## 4 Phenomenology

The main experimental constraints on our models that explain the $(g-2)_\mu$ discrepancy $\Delta a_\mu$ come from DMDD experiments and the LHC. To understand which spectra are still viable it is crucial to understand the phenomenology of them, since the experimental exclusion power

---

[4]These scenarios appear in the $(g-2)_\mu$ context for large $\mu\tan\beta$, see e.g. Ref. [84].

[5]This repository contains both the raw data and a single CSV file that summarizes the SUSY parameters, masses, and the phenomenology explained in Section 3 of all the surviving spectra. Each line in the CSV file corresponds to one particular spectrum, whose name is uniquely specified and corresponds to the names of the directories of the raw data. The contents of the CSV file is further explained in [1].

varies depending on the composition of the neutralinos and charginos. In this section, we therefore take a look at the different scenarios and contributing compositions, and describe in detail the properties of these spectra. Knowing these properties is also relevant for considering future experimental setups, e.g. for LHC studies where the exclusion power heavily depends on the assumed model.

We first discuss the DM phenomenology of the LSP. We assume that the DM abundance is determined by thermal freeze-out and require that the lightest neutralino saturates $\Omega_{DM}h^2$ with the observed value of 0.12 [9] within 0.03 to allow for a theoretical uncertainty on the relic-density calculation. As explained above, the mass eigenstate of the DM particle is a mixture of bino, wino and higgsino interaction eigenstates. To obtain the correct relic density in the pMSSM with a pure state, one can either have a higgsino with a mass of $m_{\widetilde{\chi}_1^0} \simeq 800$ GeV or a wino with $m_{\widetilde{\chi}_1^0} \simeq 2.5$ TeV. Spectra that saturate the relic density with lower DM masses necessarily are predominantly bino-like, mixed with higgsino/wino components. Negligible higgsino/wino components are found in so-called funnel regions [127, 128], i.e. regions where the mass of the DM particle is roughly half of the mass of the $Z$ boson, SM-like Higgs boson or heavy Higgs boson. In such a scenario, the mass of the neutralino can even get below 100 GeV with $M_1 < 100$ GeV, and in particular the early-universe DM annihilation cross section is enhanced for $m_{\widetilde{\chi}_1^0} \simeq m_h/2$ and $M_Z/2$. Moreover, spectra with another particle close in mass to the LSP can satisfy the relic density constraint without having a large wino/higgsino component too, due to the co-annihilation mechanism [129].

Requiring that our spectra are simultaneous minimally fine-tuned and satisfy the $\Delta a_\mu$ constraint removes two types of solutions where the DM relic density constraint is satisfied. Firstly, the case where the lightest neutralino is predominantly wino-like results in a fine-tuned spectrum: to obtain the right relic density $M_2 \simeq 2.5$ TeV for a pure wino, so $|\mu| > 2.5$ TeV in that scenario. Secondly, the pure-higgsino solutions with the right $\Omega_{DM}h^2$ do result in $\Delta_{EW} < 100$, but do not allow for an explanation of $\Delta a_\mu$, which will explicitly be shown in Section 4.4. Therefore we will see that our solutions feature predominantly bino-like LSPs. Due to the combined $\Delta a_\mu$ constraint (requiring high $\tan\beta$), DMDD limits and the FT requirement, the composition has a small higgsino component ($< 20\%$) and a negligible wino component.

On the left-hand side of Fig. 1 we show the spectra that survive all constraints and have $\Delta_{EW} < 100$. Lower values for $\Delta_{EW}$ are generally found for lower DM masses. The mass of the DM particle does not exceed 500 GeV, which is a direct result of the combined requirements of having $\Delta_{EW} < 100$ and a sufficiently high contribution to $\Delta a_\mu$. The lowest-obtained value is $\Delta_{EW} = 12.3$. From the right-hand side of Fig. 1, we can distinguish three different types of DM early-universe annihilation mechanisms: the funnel regions, the coannihilation regions and the bino-higgsino solution (indicated with $b\bar{b}$ and $t\bar{t}$). For clarity we show in Fig. 2 the same plot split out per annihilation channel, where it clearly can be seen that for example the $t\bar{t}$ and $b\bar{b}$ annihilation regimes overlap.

Before discussing the phenomenology of each of these regions in more detail, we first discuss the compositions of the LSP, the second-to-lightest neutralino and the lightest chargino. As anticipated in the previous section, and as shown in Fig. 3, we find that the LSP is predominantly bino-like and has a small higgsino component. Larger higgsino components are generally found for spectra that show larger values of $\langle \sigma v \rangle$. The second-to-lightest neutralino and the lightest chargino are either wino-like, higgsino-like, or mixed wino-higgsino states. It might be surprising to read that spectra with bino-higgsino LSPs are allowed to have wino-like $\widetilde{\chi}_2^0/\widetilde{\chi}_1^\pm$, as one would expect that in general these sparticles would be predominantly higgsino-like. Such configurations can however be found in spectra for which $M_1$, $M_2$ and $|\mu|$ are all of $\mathcal{O}(100)$ GeV with $M_2$ being smaller than $|\mu|$, and that have moderate to large values of $\tan\beta$ ($10 \lesssim \tan\beta \lesssim 20$). From Eq. (4) one may infer that for such spectra, little mixing can

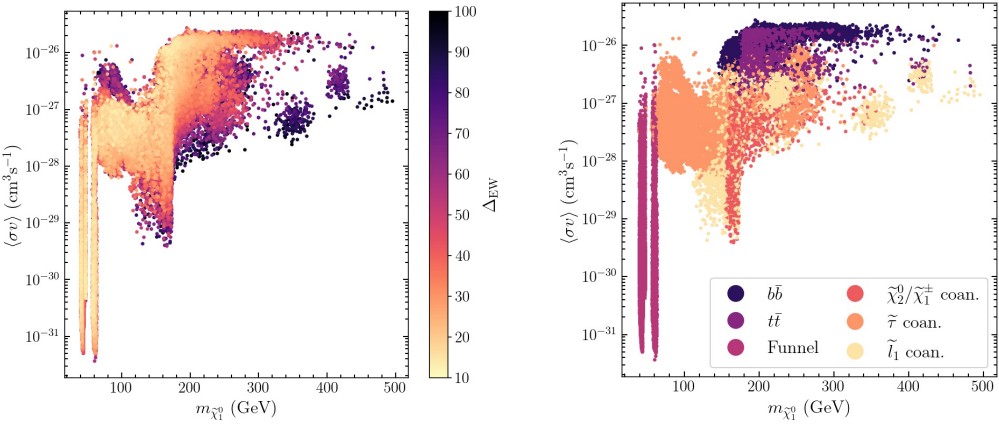

Figure 1: The mass of the DM particle ($m_{\widetilde{\chi}_1^0}$) vs the velocity-weighted annihilation cross section ($\langle\sigma v\rangle$). The value of $\Delta_{\mathrm{EW}}$ is shown as a color code on the left, where the points are ordered such that spectra with lower values of $\Delta_{\mathrm{EW}}$ lie on top of those with higher values of $\Delta_{\mathrm{EW}}$. On the right we show the dominant early-universe annihilation process that contributes to the value of $\Omega_{\mathrm{DM}}h^2$. In both plots, we only show points that satisfy all experimental constraints, and have $133\times10^{-11} < \Delta a_\mu < 369\times10^{-11}$, allowing for a $2\sigma$ uncertainty.

take place between the bino and wino. This results in negligible wino components of the LSP, whereas $\widetilde{\chi}_1^\pm$ and $\widetilde{\chi}_2^0$ can be predominantly wino-like. Moreover, decreasing $|\mu|$ for such models will not only result in a higher higgsino-component of the LSP, but counter-intuitively also in a *higher* wino component, while the wino component of $\widetilde{\chi}_1^\pm$ and $\widetilde{\chi}_2^0$ then *decreases*. The composition of the $\widetilde{\chi}_1^\pm$ and $\widetilde{\chi}_2^0$ sparticles is relevant for the LHC phenomenology, as those spectra where these are predominantly higgsino-like are typically difficult to probe at the LHC due to low production cross sections compared to the pure wino $\widetilde{\chi}_1^\pm/\widetilde{\chi}_2^0$ case.

In what follows, we will explore the DM phenomenology of each of these regimes in some more detail (Section 4.1-4.3). We also discuss their LHC phenomenology, and explain why our solutions elude the LHC constraints. This allows us to identify gaps in the LHC search program for supersymmetric particles. We end our discussion on the phenomenology of the found solutions by discussing the sensitivity of DMDD experiments in Section 4.4.

## 4.1 LHC phenomenology for the funnel regimes

We start with discussing the DM phenomenology of the funnel regions, of which there are two in our spectra [6]. The first one centers around $m_{\widetilde{\chi}_1^0} \simeq 40$ GeV, which is slightly less than $M_Z/2$. This can be explained as follows. The velocities of the DM particles were much higher in the early universe than what they are in the present-day universe. This means that DM annihilations via s-channel $Z$ exchanges could happen on-resonance in the early universe, whereas in the present-day universe these exchanges only happen off-resonance. This also explains the fact that the value for $\langle\sigma v\rangle$ is allowed to get orders of magnitude smaller than the value that one usually expects for a thermal relic (around $\langle\sigma v\rangle = 3 \cdot 10^{-26}$ cm$^3$s$^{-1}$ for a DM mass of 100 GeV). These models are characterized by small wino/higgsino components of the LSP - otherwise the early-universe annihilation would be too efficient, resulting in a too-low value of $\Omega_{\mathrm{DM}}h^2$. The second funnel region is centered around $m_{\widetilde{\chi}_1^0} \simeq 60$ GeV, slightly less than $m_h/2$. These DM particles annihilated in the early universe predominantly via s-channel SM-like Higgs exchanges. No solutions are found for spectra with DM masses in-between the two

---

[6]The heavy Higgs funnel is not identified here, and will be left for future study.

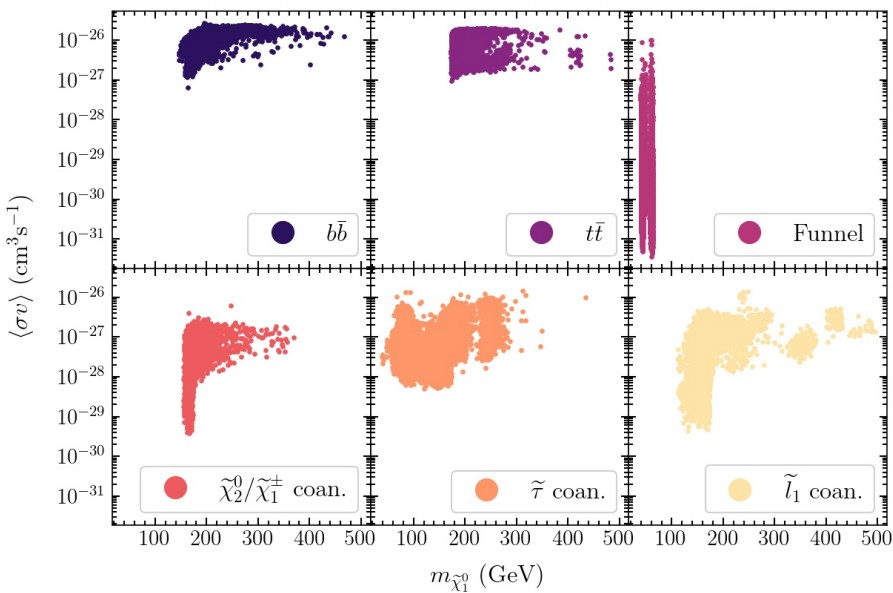

Figure 2: The mass of the DM particle ($m_{\widetilde{\chi}_1^0}$) vs the velocity-weighted annihilation cross section ($\langle \sigma v \rangle$). The same points as in Fig. 1 are shown, but split out individually for each early-universe annihilation process.

funnel regions. Here, the wino/higgsino component necessarily needs to increase to satisfy the $\Omega_{\mathrm{DM}} h^2$ requirement, and these spectra are excluded by DMDD experiments. The minimal value of $\Delta_{\mathrm{EW}}$ for these spectra is 13.2.

We now consider the compositions of $\widetilde{\chi}_1^0$, $\widetilde{\chi}_2^0$ and $\widetilde{\chi}_1^\pm$, and identify the mass difference between the LSP and the next-to-lightest SUSY particles in the funnel regimes, as this is important to understand the LHC phenomenology of these regions. The two funnel regimes are characterized by light ($m_{\widetilde{\chi}_1^0} < 100$ GeV) bino-like LSPs. The $\widetilde{\chi}_1^\pm$ and $\widetilde{\chi}_2^0$ are degenerate in mass. They are wino mixtures for masses around $100-200$ GeV, while they become higgsino-like for heavier $\widetilde{\chi}_1^\pm$/ $\widetilde{\chi}_2^0$ (up to $m_{\widetilde{\chi}_1^\pm/\widetilde{\chi}_2^0} \simeq 500$ GeV). The mass gap between $\widetilde{\chi}_1^0$ and $\widetilde{\chi}_2^0$ or $\widetilde{\chi}_1^\pm$ ($\Delta(m_{\widetilde{\chi}_2^0}, m_{\widetilde{\chi}_1^0})$ or $\Delta(m_{\widetilde{\chi}_1^\pm}, m_{\widetilde{\chi}_1^0})$) is at least around 50 GeV, and exceeds 100 GeV for $m_{\widetilde{\chi}_1^\pm} \gtrsim 150$ GeV (see Fig. 4, left panel). The masses of the sleptons are heavier than (at least) the masses of $\widetilde{\chi}_2^0$ and $\widetilde{\chi}_1^\pm$. Three different sorts of decays for $\widetilde{\chi}_2^0$ can be identified that are relevant final-state topologies for LHC searches:

1. $\widetilde{\chi}_2^0 \rightarrow h \widetilde{\chi}_1^0$ when $\Delta(m_{\widetilde{\chi}_2^0}, m_{\widetilde{\chi}_1^0}) > m_h$,

2. $\widetilde{\chi}_2^0 \rightarrow Z \widetilde{\chi}_1^0$ when $\Delta(m_{\widetilde{\chi}_2^0}, m_{\widetilde{\chi}_1^0}) > M_Z$,

3. off-shell decays when $\Delta(m_{\widetilde{\chi}_2^0}, m_{\widetilde{\chi}_1^0}) < M_Z$.

For $\widetilde{\chi}_1^\pm$, there are only two sorts of decays

1. $\widetilde{\chi}_1^\pm \rightarrow W^\pm \widetilde{\chi}_1^0$ when $\Delta(m_{\widetilde{\chi}_1^\pm}, m_{\widetilde{\chi}_1^0}) > M_W$,

2. off-shell decays when $\Delta(m_{\widetilde{\chi}_1^\pm}, m_{\widetilde{\chi}_1^0}) < M_W$.

We now determine why our points in the funnel region survive the LHC constraints. Given that the sleptons in these spectra are heavier than $\widetilde{\chi}_2^0$ and $\widetilde{\chi}_1^\pm$, searches for $\widetilde{\chi}_2^0 \widetilde{\chi}_1^\pm$ production with on-shell decays of $\widetilde{\chi}_2^0 \rightarrow Z \widetilde{\chi}_1^0$, such as those in Ref. [130–133], are most sensitive to our spectra. However, whenever $\Delta(m_{\widetilde{\chi}_2^0}, m_{\widetilde{\chi}_1^0}) > m_h$, we find that in our models there exists a mixture

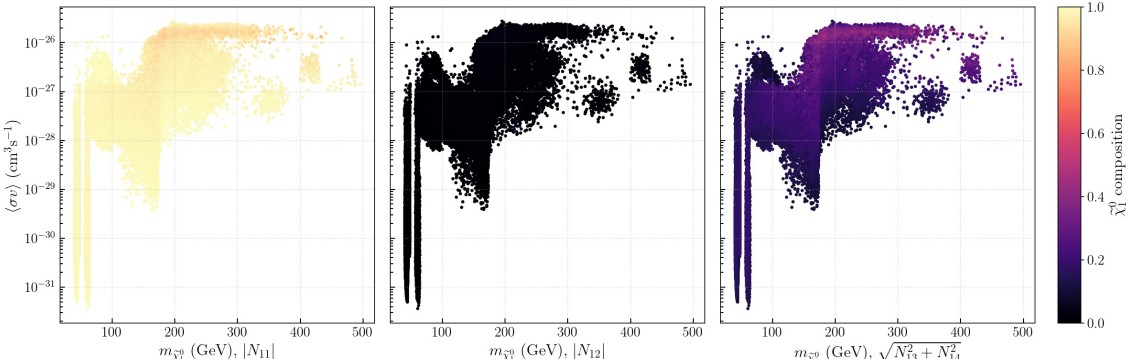

Figure 3: The mass of the DM particle ($m_{\widetilde{\chi}_1^0}$) vs the velocity-weighted annihilation cross section ($\langle\sigma v\rangle$). The composition of the LSP is shown as a color code, with the bino component $|N_{11}|$ indicated on the left, the wino component $|N_{12}|$ in the middle, and the higgsino component $\sqrt{N_{13}^2 + N_{14}^2}$ on the right.

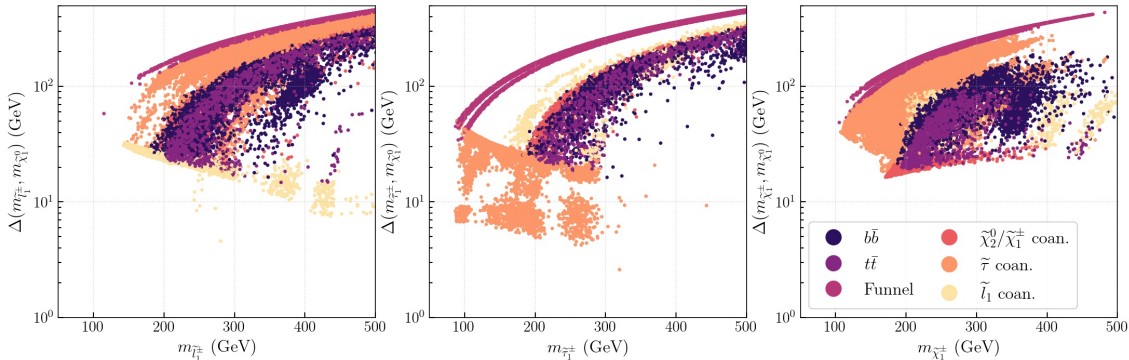

Figure 4: The mass difference between the DM particle and the lightest chargino (left), lightest smuon (middle) and lightest stau (right) versus the mass of the heavier particle. The color code represents the dominant early-universe annihilation channel.

between $\widetilde{\chi}_2^0 \to h\widetilde{\chi}_1^0$ and $\widetilde{\chi}_2^0 \to Z\widetilde{\chi}_1^0$ decays. This is part of the reason why our models evade the LHC limits: the sensitivity of the experiments drops when $\widetilde{\chi}_2^0$ can decay into the SM-like Higgs boson [131, 134]. A second reason why these spectra evade the LHC limits is that the simplified limits of the searches mentioned above assume a wino-like $\widetilde{\chi}_2^0\widetilde{\chi}_1^\pm$ pair, whereas we deal with mixed wino-higgsino pairs. To interpret the above-mentioned analyses, we show in the left panel of Fig. 5 the average cross section per 10 by 10 GeV bin for $\widetilde{\chi}_2^0\widetilde{\chi}_1^\pm$ production. We determined whether a given model point is excluded by parameterizing the upper bounds on the cross sections as shown in Ref. [132], Fig. 7 and 8, Ref. [131], Fig. 11 and Ref. [133], Fig. 5 and 6. We find that our cross sections in the regime where $M_Z < \Delta(m_{\widetilde{\chi}_2^0}, m_{\widetilde{\chi}_1^0}) < m_h$ do no not exceed the 95% confidence level (CL) limits. We expect this situation to change if more LHC data is collected, making the LHC sensitive to this part of the funnel parameter space. The models with off-shell decays are slightly more constrained by the current results of the LHC experiments. Particularly Ref. [133] excludes some of our spectra in this regime that have $m_{\widetilde{\chi}_1^\pm}$ up to 210 GeV and $\Delta(m_{\widetilde{\chi}_2^0}, m_{\widetilde{\chi}_1^0}) < 55$ GeV. These spectra are explicitly removed from the plots. The LHC shows limited sensitivity to the models in the mass range of $55$ GeV $< \Delta(m_{\widetilde{\chi}_1^\pm}, m_{\widetilde{\chi}_1^0}) < M_Z$. To gain full sensitivity to the funnel regions, this mass range is an important domain to cover in the LHC searches.

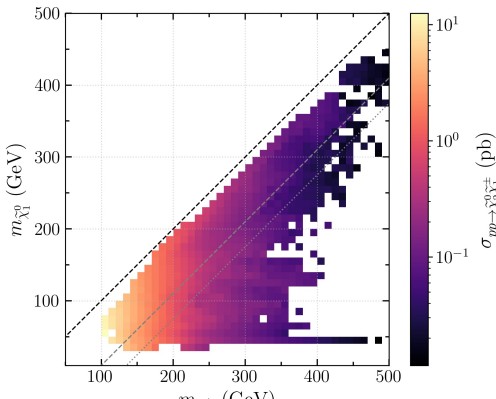 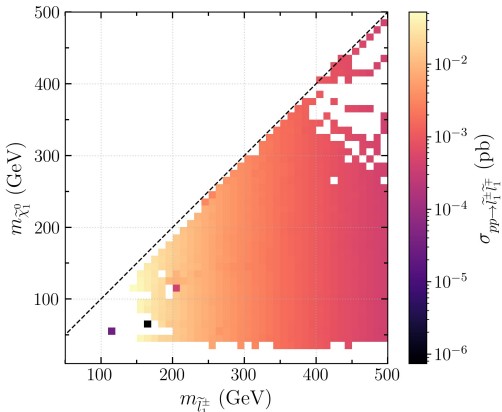

Figure 5: The mass of the DM particle versus the mass of the lightest chargino (left) and smuon (right), combined in 10 by 10 GeV bins. The average production cross section of $\sigma_{pp \to \widetilde{\chi}_2^0 \widetilde{\chi}_1^\pm}$ (left) and $\sigma_{pp \to \widetilde{l}_1^\pm \widetilde{l}_1^\mp}$ (right) is shown in color code for each bin. The dashed black line in the plot on the left-hand side shows the limit where $m_{\widetilde{\chi}_1^0} = m_{\widetilde{\chi}_1^\pm}$, whereas the gray dashed (dotted) lines show $m_{\widetilde{\chi}_1^\pm} = m_{\widetilde{\chi}_1^0} + M_Z$ ($m_{\widetilde{\chi}_1^\pm} = m_{\widetilde{\chi}_1^0} + m_h$). The dashed black line in the plot on the right-hand side shows $m_{\widetilde{\chi}_1^0} = m_{\widetilde{l}_1^\pm}$.

## 4.2 LHC phenomenology for the coannihilation regimes

The second regime is the coannihilation regime, whose DM phenomenology we now discuss. It starts to open up at DM masses of roughly 75 GeV, as no charged sparticles (and therefore no coannihilation partners other than the sneutrino) can exist with masses below 85 GeV due to the LEP/LHC bounds. Three different types of coannihilation partners are identified: first-/second-generation sleptons, third-generation sleptons, and charginos or heavier neutralinos. Interestingly, only with the help of slepton coannihilations the DM particle can have a mass between $\mathcal{O}(70-150)$ GeV and still give the right $\Omega_{\mathrm{DM}} h^2$. To obtain the right relic density in this regime without a slepton-coannihilation partner, one generally needs high higgsino fractions, which increases the value of $\sigma_{\mathrm{SI,p}}$ beyond the exclusion limit of the DMDD experiments. The lowest values of $\Delta_{\mathrm{EW}}$ are found in the stau-coannihilation regime ($\Delta_{\mathrm{EW}} = 12.3$), while the first-/second-generation slepton and chargino/neutralino regimes result in lowest values $\Delta_{\mathrm{EW}} = 14.4$ and $\Delta_{\mathrm{EW}} = 16.4$ respectively. The coannihilation regimes are all characterized by small mass differences between the LSP and its coannihilation partner(s).

The first type of coannihilation is that of first-/second-generation sleptons ($\widetilde{l}_1^\pm$). The compression between $m_{\widetilde{l}_1^\pm}$ and $m_{\widetilde{\chi}_1^0}$ is increased for higher LSP masses such that the right $\Omega_{\mathrm{DM}} h^2$ can still be obtained. By computing the production cross section (see Fig. 5), and comparing these to the results of Fig. 20 of Ref. [134], we see that spectra with $\Delta(m_{\widetilde{\chi}_2^0}, m_{\widetilde{\chi}_1^0}) > M_Z$ are under strong constraints from searches for $\widetilde{\chi}_2^0 \widetilde{\chi}_1^\pm \to \widetilde{l}\widetilde{l}l \, \nu_l$. We explicitly remove those points from our data, leaving only models with $\Delta(m_{\widetilde{\chi}_2^0}, m_{\widetilde{\chi}_1^0}) < M_Z$. The $\widetilde{\chi}_1^\pm$ and $\widetilde{\chi}_2^0$ sparticles of the surviving models are typically higgsino-like with a small wino component, and have masses between 180 and 500 GeV.

The second coannihilation regime is characterized by low $\widetilde{\tau}_1^\pm$ masses. The masses of $\widetilde{\chi}_1^\pm / \widetilde{\chi}_2^0$ can still be as light as 105 GeV in this regime, where they are predominantly wino-like. The higgsino component of these particles increases when their masses increase, up to $m_{\widetilde{\chi}_1^\pm / \widetilde{\chi}_2^0} \simeq 500$ GeV. Although we have a large production cross section for the wino-like $\widetilde{\chi}_1^\pm / \widetilde{\chi}_2^0$ pair, these models are not constrained by the LHC experiments due to the presence of the light staus. The staus are often lighter than $\widetilde{\chi}_1^\pm$ and $\widetilde{\chi}_2^0$, and the searches for $\widetilde{\tau}_1^\pm$-mediated decays of $\widetilde{\chi}_1^+ \widetilde{\chi}_1^- / \widetilde{\chi}_1^\pm \widetilde{\chi}_2^0$ production have no sensitivity when $\Delta(m_{\widetilde{\chi}_1^0}, m_{\widetilde{\tau}_1^\pm}) < 100$ GeV [135,136]. The

latter holds for our spectra in the second coannihilation regime, since the mass differences between the LSP and $\tilde{\tau}_1^\pm$ are between $5-50$ GeV in that case. Additionally, relatively few LHC searches for low-mass $\tilde{\tau}^\pm$ particles exist. Small $\tilde{\tau}^+\tilde{\tau}^-$ production cross sections and low signal acceptances make these searches difficult, so the experiments have no constraining power in the compressed regime [137, 138]. *We suggest a dedicated low mass $\tilde{\tau}^\pm$ search without an assumed mass degeneracy between $\tilde{\tau}_1^\pm$ and $\tilde{\tau}_2^\pm$ to probe the sensitivity of the LHC to these scenarios.* The last coannihilation regime has a $\tilde{\chi}_1^\pm$ or $\tilde{\chi}_2^0$ that is close in mass to the LSP. Interestingly, although the mass compression for the slepton coannihilation regimes needs to increase to obtain the right relic density for higher DM masses, for the gaugino-coannihilation regime it needs to decrease instead. Regarding the LHC phenomenology, note that although the slepton masses in these regions can be $\mathcal{O}(200)$ GeV, the results from the $\tilde{l}_{R,L}^+ \tilde{l}_{R,L}^-$ searches with $\tilde{l}^\pm = \tilde{e}^\pm, \tilde{\mu}^\pm$ or $\tilde{\tau}^\pm$ (e.g. [138–140]) are not directly applicable here, as often one or more of the chargino/heavier neutralino states is lighter than the sleptons. Therefore, the slepton will not decay with a 100% branching ratio to $\tilde{\chi}_1^0 l^\pm$, although this is assumed in the above-mentioned searches. Instead, in this regime, only the $\tilde{\chi}_1^\pm \tilde{\chi}_2^0$ searches are of relevance, similar to the case in the funnel region discussed above. The mass compression between the LSP and wino-higgsino like $\tilde{\chi}_1^\pm / \tilde{\chi}_2^0$ sparticles is generally around 15-20 GeV, and Ref. [133] excludes our solutions with $m_{\tilde{\chi}_1^\pm}$ up to $140-180$ GeV.

## 4.3 LHC phenomenology for the bino-higgsino LSP

The last regime we identify consists of bino-higgsino LSPs and is labeled with $b\bar{b}$ and $t\bar{t}$. These early-universe annihilation channels are mediated by either s-channel $Z$ or $h/H$ exchanges. The $t\bar{t}$ annihilation channel opens up when $m_{\tilde{\chi}_1^0}$ becomes larger than the mass of the top quark $m_t$, as then the invariant mass of the two LSPs is enough to create a $t\bar{t}$ pair [7]. For the $Z$-exchange channel this annihilation becomes favored over the annihilation into a lighter fermion pair, since any $Z$-mediated annihilation of two Majorana fermions is helicity suppressed at tree level [141]. This is explained as follows. The two identical LSPs form a Majorana pair. Such a pair is even under the operation of charge-conjugation $C = (-1)^{L+S}$ with $S$ the total spin and $L$ the total orbital angular momentum, so $L$ and $S$ must either both be even, or both be odd. Taking the limit of zero velocity, as the present-day velocity of DM particles is non-relativistic, we may assume $L = 0$ and even $S$. The final-state fermion pair can have a total spin of $S = 1$ or $S = 0$, but only the latter is allowed for the Majorana-pair annihilation in the non-relativistic limit. For a Dirac-field pair, an $S = 0$ configuration is obtained if the fermion and anti-fermion are from different Weyl spinors: a left- and right-handed one. In the SM, a coupling with this combination only arises (at tree level) by a mass insertion. Therefore, the transition amplitude is proportional to the mass of the final-state fermions, and a decay to a heavier pair of fermions is generally preferred. In spectra where $\tan\beta$ is large we also see the heavy-Higgs-mediated decays to $b\bar{b}$, as the bottom-Yukawa coupling is enhanced. As can be seen in Fig. 4, in the regime of $m_{\tilde{\chi}_1^0} \gtrsim m_t$, the masses of $\tilde{\chi}_1^\pm$ and $\tilde{\chi}_2^0$ are relatively close to that of the LSP, so due to the coannihilation mechanism these spectra tend to show slightly lower values of $\langle \sigma v \rangle$ than naively would be expected.

The minimal value of $\Delta_{\text{EW}}$ is around 14.2 for these models. The $\tilde{\chi}_2^0$ and $\tilde{\chi}_1^\pm$ are predominantly higgsino-like with masses from 180 to 500 GeV. Due to their small production cross section, the LHC searches do not have exclusion power in this regime.

---

[7]The annihilation to a $W^+W^-$ pair is possible when $m_{\tilde{\chi}_1^0} > M_W$. However, this is constrained by DMDD due to the high wino/higgsino fraction that is necessary for this channel.

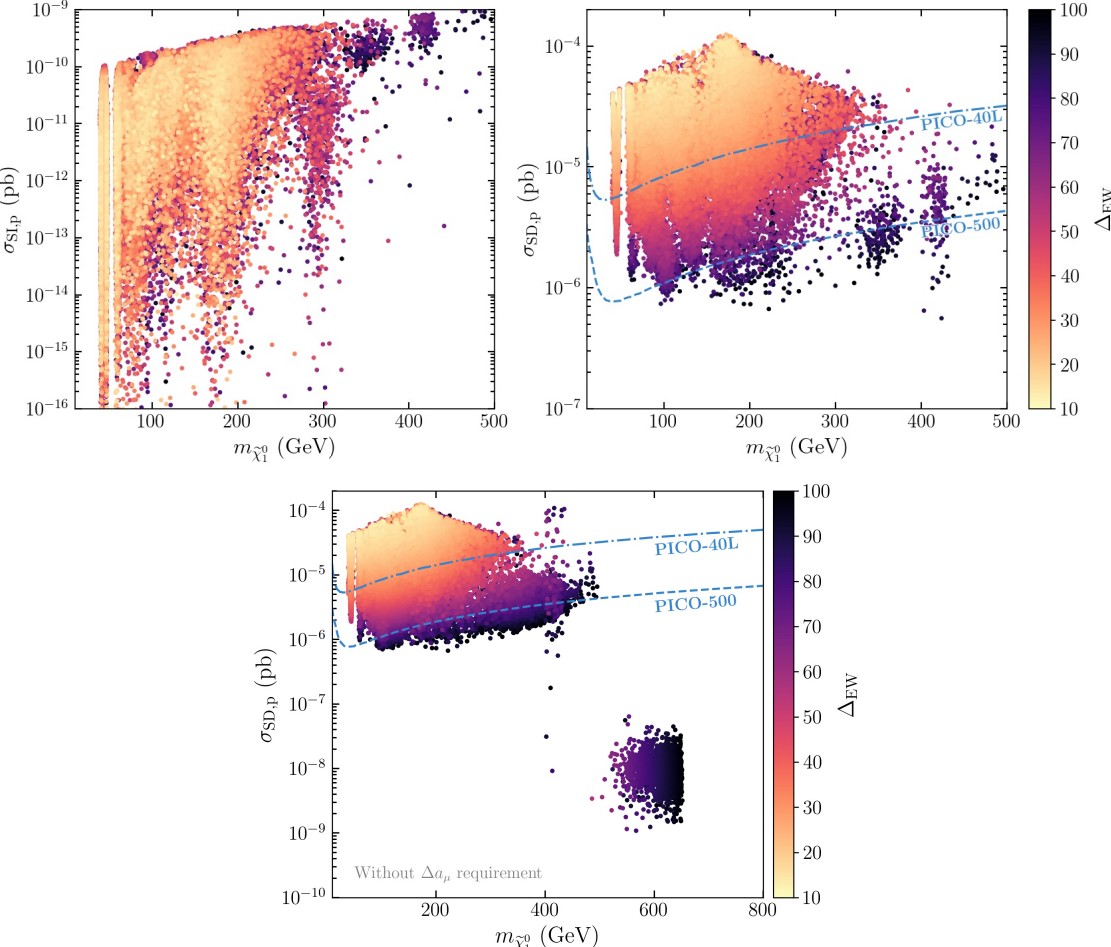

Figure 6: Top right (left): The mass of the DM particle versus the spin-(in)dependent cross section $\sigma_{\mathrm{SD,p}}$ ($\sigma_{\mathrm{SI,p}}$). The value of $\Delta_{\mathrm{EW}}$ is shown in color code. We also show the projected PICO-40L and PICO-500 central limits on $\sigma_{\mathrm{SD,p}}$ [142]. The points are ordered such that those with lower values of $\Delta_{\mathrm{EW}}$ lie on top of those with higher values. Bottom: The mass of the DM particle versus $\sigma_{\mathrm{SD,p}}$ for spectra satisfying all constraints listed in Section 3 except the $\Delta a_{\mu}$ requirement. This plot contains the data of the present study combined with that from Ref. [64], where the requirement on $a_{\mu}$ was not taken into account.

## 4.4 Dark-matter direct detection experiments

In the previous subsections we discussed the phenomenology of the viable spectra at the LHC. We now comment on the sensitivity of DMDD experiments. We have seen that the LSP in our spectra is always bino-like with a small higgsino component (Fig. 3). We find that the relative size of the wino component of the LSP is constraint by DMDD experiments: higher wino components result in larger values of $\sigma_{\mathrm{SI,p}}$ and $\sigma_{\mathrm{SD,p}}$. Surprisingly, this indirectly also places a lower bound on $|\mu|$: decreasing $|\mu|$ for our models will not only result in a higher higgsino-component, but also in a higher wino component of the LSP, as more mixing between the wino and bino components is then allowed. Therefore, decreasing $|\mu|$ for these scenarios is limited by the constraints imposed by the DMDD experiments.

The resulting values for $\sigma_{\mathrm{SI,p}}$ and $\sigma_{\mathrm{SD,p}}$ of the surviving models may be seen in Fig. 6. While the value of $\sigma_{\mathrm{SI,p}}$ varies by over 7 orders of magnitude, $\sigma_{\mathrm{SD,p}}$ is relatively constrained. We moreover observe that $\sigma_{\mathrm{SD,p}}$ is directly correlated with $\Delta_{\mathrm{EW}}$: lower values of $\sigma_{\mathrm{SD,p}}$ result in

higher values of $\Delta_{EW}$. The value of $\sigma_{SD,p}$ decreases with smaller higgsino fractions in the LSP, while for a given fixed LSP mass $\Delta_{EW}$ increases since $|\mu|$ needs to increase. In this figure we also indicate the projected limit of the PICO-40L and the PICO-500 experiments [142]. We observe that the latter one is sensitive to all of our solutions with $\Delta_{EW} < 62$. The LUX-ZEPLIN experiment [143] (whose projected limit is not shown in Fig. 6) will probe all of our solutions with $\Delta_{EW} < 100$.

This shows an important message, namely that *future DMDD experiments that probe $\sigma_{SD,p}$ will be sensitive to all our solutions, irrespective of the masses and compositions of the rest of the sparticle spectrum.* That the $\Delta a_\mu$ requirement is crucial to obtain this conclusion is shown in the bottom panel of Fig. 6, where we show both the spectra from this work and those from Ref. [64] without imposing the $\Delta a_\mu$ constraint. One may observe that in this case spectra survive with $m_{\tilde{\chi}_1^0} > 500$ GeV that show very small values of $\sigma_{SD,p}$. These pure higgsino solutions have vanishing couplings to the $Z$-boson and therefore evade detection at future DMDD experiments, but do not satisfy the $\Delta a_\mu$ requirement.

## 5  Conclusion

In this paper we for the first time have analyzed the spectra in the pMSSM that are minimally fine-tuned, result in the right $\Omega_{DM}h^2$ *and* simultaneously offer an explanation for $\Delta a_\mu$. We make these spectra publicly available under [1].

In terms of DM phenomenology, we have distinguished three interesting branches of solutions: the funnel regimes, three types of coannihilation regimes, and the generic bino-higgsino solution. All these solutions have in common that the LSP is predominantly bino-like with a small higgsino component. Masses of the DM particle range between $39 - 495$ GeV. We discussed the phenomenology at the LHC for each of the regimes. The first and second regime are relatively more constrained by $\tilde{\chi}_2^0\tilde{\chi}_1^\pm$ searches at the LHC than the last regime, which is due to the lower wino-components and higher masses of the $\tilde{\chi}_2^0/\tilde{\chi}_1^\pm$ sparticles that is typical in the last regime. On the other hand, in particular when the coannihilation partner of the LSP is a light stau, the LHC searches show little to no sensitivity to our found solutions. Our solutions motivate further the ongoing efforts at the LHC to probe pMSSM spectra that feature (compressed) higgsino-like production of $\tilde{\chi}_2^0\tilde{\chi}_1^\pm$ pairs. In addition, to increase the sensitivity of the LHC to our found solutions, we find that a dedicated low-mass $\tilde{\tau}^\pm$ search without an assumed mass degeneracy between $\tilde{\tau}_1^\pm$ and $\tilde{\tau}_2^\pm$ would be needed, but also that the mass-gap region of $55$ GeV $< \Delta(m_{\tilde{\chi}_2^0}, m_{\tilde{\chi}_1^0}) < M_Z$ is not probed at the LHC. Proposing a dedicated search for these regimes, however, lies beyond the scope of this work.

We find that DMDD experiments that probe $\sigma_{SD,p}$ will ultimately be sensitive to all of our minimally fine-tuned spectra. The requirement of satisfying $\Delta a_\mu$ is crucial to arive at this conclusion. This requirement excludes models with a higher-mass higgsino with $m_{\tilde{\chi}_1^0} = 550-650$ GeV as the LSP, and these spectra would evade detection by future DMDD experiments.

## Acknowledgments

MvB acknowledges support from the Science and Technology Facilities Council (grant number ST/T000864/1).

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
