# Peer review of "Dark matter, fine-tuning and (g-2)_{\mu} in the pMSSM"

_SciPost Physics, doi:SciPost Phys. 11, 049 (2021)_

## Round 1 · Referee Report · Anonymous (Referee 1) · 2021-6-2

Strengths

  1. The authors did an analysis which seems to me very complicated and time-consuming.
  2. The analysis procedure seems rigid and trustworthy.

Weaknesses

  1. The accomplishment of this work is not clearly written.
  2. The manuscript is not well organized, or at least not reader-friendly.
  3. Figures might be misleading.

Report

This work focuses on MSSM model points that solve the muon g-2 anomaly and provides the correct relic density of the dark matter. The authors also require that the models should have small fine-tuning, which is measured by Eq. (2), as well as pass the constraints from experiments. The model points passing all these criteria are shown in their figures.

This work is distinctive because, as far as I know, and as the authors stated in Sec. 1, this is the first study in which all these constraints are taken into account in drawing such scatter plots, i.e., with correct relic density, correct muon g-2, small fine-tuning, collider constraints, and DM direct detection constraints. However, although the authors did careful analyses and obtained good data, the discussion and the way of presentation are not reader-friendly or well-organized, as described below and therefore I ask for a major revision. In particular, I would ask to solve the following issue 1) before further consideration.

1) The biggest concern of mine is that I am not sure what is the main message of this work, or in other words, if this manuscript provides a significant progress to this field. That is, getting the data points itself counts as a significant progress only if the data points are made public; otherwise, the authors are responsible to describe and discuss the model points and clarify what the readers can learn. However, such statements are not found in Conclusion. Rather, it seems to me that the information in Conclusion has been known previously, in particular, in the authors' previous work 1612.06333. I thus ask for the authors to clarify what is the main message of this work; e.g., the progress/difference they accomplished compared to 1612.06333.

2) The authors discuss the collider phenomenology of the model points that are not excluded by colliders, but I am not sure its purpose. For example, in L258-278, the authors discuss LHC phenomenology on the model points that we already know have eluded the LHC constraints. If to characterize the property of surviving points, then the points should be compared with the excluded points. If to discuss future prospects, then the discussion should be done quantitatively with the published information on, e.g., HL-LHC prospects. It seems to me that, unfortunately, the collider phenomenology part is just a display of what they know on the points, and not leading the readers to any goals.

3) The categorization of Fig. 1 Right might be misleading. In Fig. 1 Right and others, the authors categorize the model points by the mechanism for DM relic density. However, the way it is displayed seems to me very confusing; namely, novice young (or busy senior) readers may misunderstand that the (200GeV, $10^{-26}$cm^3/s) regions are not by bbbar but only by ttbar. I recommend the authors to think again on the way of presentation.

4) Several minor comments/questions. - Which order is the g-2 calculated at? The tanbeta resummation of GM2Calc is turned on or not? - Which scale the MSSM parameters are specified at? GUT scale with the running considered, or MSUSY? - The paragraph L209-231 seems important but not well-organized. The authors should consider refine it, e.g., for the statement in L214, it is helpful if the authors could provide a plot like Fig. 1 but with the color showing the magnitude of bino-component ($N_{11}$). - L272: The method of recasting should be detailed. - L297: What happened to "those points"? Are they rejected by some of the above-given criteria, or the authors just assume that the points have been excluded?

---

## Round 2 · Referee Report · Anonymous (Referee 1) · 2021-6-24

Report

The revision by the authors improves the visibility and clarity of the manuscript and sufficiently answers my previous comments 3) and 4). In particular, the updated Fig. 2 constitutes an interesting observation.
However, this revision together with the authors' comment does not alleviate my main concern 1) as described below.

According to the manuscript and the comment, the following strengths of this work are reported.
A) A dedicated numerical study is performed and the data are made public on Zenodo.
B) Spin-dependent dark matter searches are found crucial for the natural model points of the MSSM with correct dark matter abundance.
C) It shows that a dedicated search for low-mass staus is required.
These points are clearly summarized in Sec. 5 of the manuscript.

I agree that the authors have accomplished these observations and they are transparently described in the text and figures.
In particular, A) is a great accomplishment of this work and possibly enough to make the manuscript eligible for publication.

The claim B) was already known in the authors' previous work 1612.06333 (or works by other authors 1609.06735).
What is new in this work is, as the authors wrote in the comment, these spectra include solutions for the muon g-2 anomaly.
Namely, 1612.06333 reported that DMDD-SD is useful to search for natural models, while this work reports DMDD-SD is useful to search for natural models that solve the muon g-2 anomaly.
However, because natural models include natural points that solve g-2, the new claim B) is already included in the claim of 1612.06333.
In other words, it is already obvious in 1612.06333 that natural g-2 models will be searched for by DMDD-SD because the muon g-2 anomaly, as well as the naturalness, favors lighter SUSY particles.
Consequently, I will not regard this B) as a new accomplishment of this work.

Furthermore, the authors' claim C) sounds problematic.
I feel, by this claim and the other discussion in Sec.4, that the authors might possibly confuse the premise and the result.
The authors wrote in the paragraph (line 348) that low-mass stau searches are required to probe the unexplored points with stau coannihilation because these points are less constrained than the other models.
However, everyone knows the importance; all LHC experimentalists working for SUSY searches know that direct stau searches are necessary to probe stau coannihilation.
Stau coannihilation models are not explored, not because they do not work for staus, but because the direct stau searches are very difficult compared to sleptons because of the difficulties in tau identification and missing-momentum reconstruction.
The statement in line 359, which is made without acknowledging the difficulty, looks very strange to me, and unfortunately I feel that the authors may have some misunderstanding on the LHC SUSY searches.

The problem in C) is related to my previous comment 2), which is not resolved by this revision.
The collider phenomenology part is not improved sufficiently.

To assess the manuscript, I compare the merit obtained by A) with the problems I found in C).
I conclude that it is difficult to recommend this manuscript to the Editor, and do not ask for further revisions because this revision does not sufficiently improve the discussion on collider phenomenology.
  • validity: low
  • significance: ok
  • originality: good
  • clarity: good
  • formatting: perfect
  • grammar: perfect

Author:  Melissa van Beekveld  on 2021-07-01  [id 1538]

(in reply to Report 1 on 2021-06-24)
Category:
objection

We are surprised to see a negative recommendation from the referee. We do not agree with the referee's concern on point B) and C), and would like to explain our concerns below.

Firstly, the referee claims that 'The claim B) was already known in the authors' previous work 1612.06333'. Here we show that in fact this does not follow from our previous work, and that the inclusion of $\Delta a_{\mu}$ is crucial to reach our conclusion. In 1612.06333, we considered points with $\Delta_{\rm EW}<10$, whereas in this work we consider $\Delta_{\rm EW}<100$ and in fact no points with $\Delta_{\rm EW}<10$ have survived at all. We may analyse the data of 1612.06333 using the constraint $\Delta_{\rm EW}<100$ (see the attached figure), and examine whether the conclusion remains unchanged. What one observes in the figure is that \emph{not} all spectra with low fine-tuning can be probed by future DMDD experiments. This happens because pure higgsino solutions are allowed if one only uses the requirement of $\Delta_{\rm EW}<100$ (left-hand side of the figure). However, if one includes the $\Delta a_{\mu}$ constraint (right-hand side of the figure), these pure higgsino solutions disappear. This happens since $|\mu|$ cannot be too high to satisfy $\Delta a_{\mu}$ (for that, $|\mu| < 500 $ GeV). This shows that the conclusion of the current paper certainly does not follow from our previous work, and that the referee's argument on point B) is invalid.

Secondly, regarding point C), we feel that a few sentences in our discussion of the LHC phenomenology are considered a weighty argument against the entire paper. The referee states that 'The statement in line 359, which is made without acknowledging the difficulty, looks very strange to me, and unfortunately I feel that the authors may have some misunderstanding on the LHC SUSY searches'. However, we mention the difficulty of such searches explicitly in our manuscript, line 357: 'Additionally, relatively few LHC searches for low-mass $\widetilde{\tau}^{\pm}$ particles exist. Small $\widetilde{\tau}^+\widetilde{\tau}^-$ production cross sections and low signal acceptances make these searches difficult, so the experiments have no constraining power in the compressed regime'. We fully understand that 1) the trigger poses a great limitation, since it puts a large $p_T$ requirement on the $\tau$, and that 2) the missing-transverse energy is reconstructed with a low resolution. We would be willing to elaborate on this point further in our manuscript, thereby addressing the referee's concern on this point. We disagree with the referee's statement that 'all LHC experimentalists working for SUSY searches know that direct stau searches are necessary to probe stau coannihilation'. That is, of course the stau coannihilation region was known previous to our study, but it was unknown that this region plays a hugely important role in the models that are low-finetuned, give the right $\Omega h^2$, and the right value of $\Delta a_{\mu}$. One of the conclusions in our work is to highlight this region in this respect.

Finally, it seems the referee has changed opinion in between the two reports, resulting in contradicting statements. The referee clearly states in the first report that 'getting the data points itself counts as a significant progress only if the data points are made public'. We have made our 15 GB-sized data-set public, and in the past week it has been downloaded 14 times. In this regard, we are also surprised to see that the validity of our paper is considered to be low: since the data is public, our analysis can be checked in full. In the second report, the referee still acknowledges the merit of this public dataset: 'A) is a great accomplishment of this work and possibly enough to make the manuscript eligible for publication.', but this is later on dismissed when compared to our discussion about LHC phenomenology.

Attachment:

---

## Round 2 · Referee Report · Anonymous (Referee 1) · 2021-7-1

Report

Dear Editor,

I am very sorry that I wrote the previous report by at least one misunderstanding regarding the item (B), as stated in the authors' reply.

As the authors wrote, 1612.06333 assumes deltaEW < 10, which is very strict assumption, while this manuscript assumes deltaEW < 100, which is a nowadays well-accepted, or popular, criterion, and the findings that a similar conclusion is obtained with deltaEW < 100 because of the g-2 is definitely very interesting.

Therefore, I have to withdraw my discussion B) in the last report and instead respect the importance on B). Accordingly, I would like to recommend the manuscript for submission because of the two importance A) and B), even with my concerns on C).

I have several proposal for some minor modifications before the submission, such as to state the difference between 1612.06333 as the authors wrote in the latest reply. In case I overlooked such statements, I would like to read the manuscript again and write to the Editor my proposal within a few days.
  • validity: good
  • significance: high
  • originality: good
  • clarity: low
  • formatting: perfect
  • grammar: perfect

Author:  Melissa van Beekveld  on 2021-07-30  [id 1627]

(in reply to Report 2 on 2021-07-01)

We thank the referee for their response, and have incorporated their suggestions in our paper as follows

1 and 2: We have rewritten lines 222-231. In section 4.4, figure 6, we have included the plot containing data from our previous work and show that the g-2 requirement forbids natural solutions with mDM > 500 GeV, and stressed in the text (line 417-424) that the g-2 requirement is crucial for our results.  We also rewrote the last paragraph of our conclusion to clarify the message of our paper (line 445-449). 

3 We have included a footnote on page 6 to offer some guidelines to the repository.

---

## Round 2 · Referee Report · Anonymous (Referee 1) · 2021-7-4

Report

Dear Editor,

Let me first apologize my misunderstanding on the previous report. The paper is reported to have importance:

A) A dedicated numerical study is performed and the data are made public on Zenodo. B) Spin-dependent dark matter searches are found crucial for the natural model points of the MSSM with correct dark matter abundance. C) It shows that a dedicated search for low-mass staus is required.

I highly evaluate the importance A), while I regard the discussion on C) in the manuscript is mainly a repetition of well-known results and observations. If A) were the sole importance of this manuscript (as I misunderstood), I believe the manuscript should be written with its main focus on the method, techniques, and usage of the data files (such as the data file format). While the manuscript focuses on phenomenology or interpretation of the data rather than data themselves, for which I proposed to reject the manuscript.

However, as written in the authors' reply, I wrongly neglected the importance B); in their previous work 1612.06333 they required the spectra be very natural (delta-EW < 10), which I feel much more strict than our standard, while here the requirement is relaxed to < 100, a very popular criterion on the naturalness. The observation in Fig. 5 that the model points with delta-EW < 90 will be explored by PICO-500 is thus of interests of other readers, and the manuscript is worth for publication with its focus on phenomenology.

In summary, the importance of this work is clearly observed in the latest reply of the authors, which resolves my biggest concern, i.e., if this manuscript provides a significant progress to this field, while I suppose that their importance is understated in the manuscript. I have therefore several minor proposals for improvement, but 2) and 3) are optional.

1) The authors should clarify the difference from 1612.06333 and what is the new observations, otherwise readers may misunderstand that this work is just an anomaly-hunters' repetition of 1612.06333. 2) (related to 1) The plot attached in the authors' objection is very interesting; in fact, it seems to me the most important plot because it is manifested that the muon g-2 anomaly will play a very important role in the DMDD experiments together with the naturalness; it will be a very nice message to the DMDD community. The authors should consider including a similar plot in the manuscript or emphasizing this observation more in, e.g., the conclusion. 3) The authors may consider to include more technical information on the data files to help readers who want to use them in the manuscript or on Zenodo.

I believe that, after a minor improvement, I will be able to recommend the manuscript for publication.

---

## Round 2 · Author Response

We've improved the clarity on the results of our paper in a number of directions:

1)

  • We have reworded the abstract slightly, stressing that we for the first time combine $g-2$ with DM properties and fine-tuning and stress our main result: DM direct detection experiments sensitive to $\sigma_{\rm SD,p}$ will probe all of our found solutions.
  • In terms of LHC phenomenology, and besides from the ongoing efforts to probe the higgsino-like production of $\widetilde{\chi}^{\pm}_1\widetilde{\chi}^0_2$ pairs, our paper shows that a dedicated search for low-mass staus is required to probe our solutions. We've adjusted the second paragraph of our conclusions accordingly.
  • In terms of DMDD phenomenology, we find that DMDD experiments sensitive to $\sigma_{\rm SD}$ probe our solutions, which is a direct result of the upper limit of $|\mu|$ imposed by the combined fine-tuning and $g-2$ constraint. Indeed, this was known in our previous work. However, what was unknown in our previous work is that these spectra would include solutions that simultaneously also satisfy the $g-2$ requirement. We've adjusted the last paragraph of our conclusion accordingly.
  • We've improved the discussion of the results in section 4 (see `several minor comments/questions, point 3).

In addition, we have published the data set, and apologize for not having done so before. We've indicated the link to the Zenodo web page in the new version of our paper.

2)

Our aim was to contrast surviving points with those that are excluded, in order to identify gaps in the search coverage of the LHC experiments. In other words: we want to understand why these spectra have eluded the LHC constraints. This is also why we discuss so extensively the compositions of the lightest chargino/second-to-lightest neutralino. This careful analysis also allows us to identify that the low-mass stau region is relatively uncovered at the LHC, but also that the mass-gap region of $55~\text{GeV} < \Delta(m_{\widetilde{\chi}^0_2},m_{\widetilde{\chi}^0_1}) < M_Z$ is not probed at the LHC. We identify these gaps, but proposing a dedicated search for these scenarios lies beyond the scope of this work.

3)

We have included a plot for Fig. 1 where we show the separate regions. We have chosen not to do the same for Fig. 4 (the other one where this might be relevant), because in that plot the relevant information is clearly visible: in the left-hand plot one can see that the `yellow' points show the mass compression (this is the slepton coannihilation regime), in the middle-plot we see that the stau coannihliation regimes shows the mass compression, and in the right-hand plot we see that the lightest chargino/next-to-lightest neutralino is compressed with the LSP.

4)

  • We determine g-2 including the two-loop corrections and turn on the $\tan\beta$ resummation (added line 189-190).
  • We specify the MSSM parameters at the SUSY scale (added line 77)
  • We improved the clarity of the text by restructuring the paragraphs, moving some of it to other sections, and added a figure that shows the composition of the LSP.
  • We slightly abused the language here. Actually, we did not do a proper recasting of the analysis, which lies beyond the scope of this paper. Instead, we computed the cross sections at NLO for each of our models using prospino, and compared with the experimental results of Ref. [132], Fig. 7 and 8, Ref. [131], Fig. 11 and Ref. [133], Fig. 5 (slepton mediated decays) and 6, which are the most relevant to our models, by making a course-grained grid of the data presented in those plots. If a cross section of a given point exceeds the upper limit on the cross section given in those plots, we excluded the point. We've added some clarifying text around line 322.
  • Using the same method of computing the cross sections, and comparing to the results of Ref. [134], Fig. 20, we have taken a conservative approach and excluded them. We've added a clarification around line 351.

---

## Round 2 · List of Changes

Brief summary of the changes:

  • reworded the abstract
  • extended the discussion of the results
  • published the dataset
  • added a figure where we show the LSP composition of the surviving models
  • added a figure to show the separate regions for the annihilation mechanisms
  • detailed the method of interpreting the experimental results of ATLAS and CMS
  • clarification of the conclusions See the author comments for more details.

---

## Round 3 · Author Response

We thank the referee for their response, and have incorporated their suggestions in our paper as follows

1 and 2: We have rewritten lines 222-231. In section 4.4, figure 6, we have included the plot containing data from our previous work and show that the g-2 requirement forbids natural solutions with mDM > 500 GeV, and stressed in the text (line 417-424) that the g-2 requirement is crucial for our results. We also rewrote the last paragraph of our conclusion to clarify the message of our paper (line 445-449).

3 We have included a footnote on page 6 to offer some guidelines to the repository.

---

## Round 3 · List of Changes

• We have included a footnote on page 6 offering guidelines to use the data
  • We have rewritten lines 222-231
  • In section 4.4, figure 6, we have included the plot containing data from our previous work and show that the g-2 requirement forbids natural solutions with mDM > 500 GeV
  • We stressed (line 417-424) that the g-2 requirement is crucial to obtain our results
  • We rewrote the last paragraph of our conclusion to clarify the message of our paper (line 445-449).

---

## Editorial Decision

published